# The Impact of Environmental Enrichment on the Cortisol Level of Shelter Cats

**DOI:** 10.3390/ani14091392

**Published:** 2024-05-06

**Authors:** Justyna Wojtaś, Piotr Czyżowski, Kamila Kaszycka, Klaudia Kaliszyk, Mirosław Karpiński

**Affiliations:** Department of Animal Ethology and Wildlife Management, University of Life Sciences in Lublin, Akademicka 13, 20-950 Lublin, Poland; justyna.wojtas@up.lublin.pl (J.W.); kamila.kaszycka@up.lublin.pl (K.K.); klaudia.kaliszyk@gmail.com (K.K.); miroslaw.karpinski@up.lublin.pl (M.K.)

**Keywords:** feline, stress, hair, environmental resources

## Abstract

**Simple Summary:**

A shelter for homeless animals is a highly stressful place for cats. This study aimed to assess whether enriching the living environment of these cats with additional resources, such as scratching posts and hiding places, would reduce long-term stress. One hundred and seventy-nine cats took part in the study. The research material consisted of hair. Cortisol levels were analyzed. The results confirmed that cats from a more enriched environment had almost half the cortisol level in their hair than those with fewer resources.

**Abstract:**

Enriching cats’ living environment in shelters is crucial in reducing their stress. Easier access to resources allows cats to display natural behavior. This study aimed to assess whether cats staying in an enriched environment would be less stressed than cats staying in a standard environment. The first group consisted of cats living in an environment with fewer resources (standard environment)—103 cats. The second group consisted of cats living in an enriched environment—76 cats. The research material consisted of hair collected to determine the cortisol level. The results indicate that cats from a more enriched environment have almost half the level of cortisol in hair than cats from an environment with fewer resources (0.059 ng/mg vs. 0.101 ng/mg; *p* = 0.000001).

## 1. Introduction

In a shelter for homeless animals, cats are exposed to many stressors. The stressors might be a large number of animals in a small area, increased infectious pressure [1,2], noise [3], veterinary treatment, or a small amount of environmental resources for a large number of cats [4]. Enriching cats’ living environment in shelters is essential in improving their well-being and preventing, limiting, and eliminating undesirable behaviors and behavioral problems [5].

The artificial and poor environment of captivity disrupts the natural repertoire of behaviors of felids, resulting in abnormal behaviors and physiological disorders. In their research, Damasceno & Genaro [6] suggest that the amount of environmental enrichment for animals living in large groups in captivity should be constantly increased, facilitating their access to resources. Easier access to resources (especially food enrichment) allows felines to display natural feeding behavior. Not only the amount of enrichment but also what types of enrichment should be introduced for shelter cats, depending on, e.g., the time they stay in a given place, may also be important [7].

Environmental enrichment for cats can be divided into five primary groups—physical resources (space), nutritional resources, elimination resources, social resources, and behavioral resources [8]. When keeping many cats, the rooms should allow the cats to maintain a distance of at least 3 m from each other [9]. However, the quality of space is more important than its quantity. Room features should include vertical structures for climbing, high places for observing the surroundings, places for resting and sleeping (both on a raised platform and the floor), scratching surfaces, and litter boxes [10].

Cortisol is a well-known and often-used physiological indicator because it allows us to assess the hypothalamic–pituitary–adrenal axis activation [11]. Hair cortisol concentration (HCC) is hypothesized to be a retrospective marker of integrated cortisol secretion and stress over more extended periods. HCC measurement for stress assessment is readily used in pets due to its easy and minimally invasive sampling procedure and the representation of more extended periods in one sample [12]. The level of cortisol in cats was examined, among others, in the works of Wojtaś (2023) [13] and Contreras (2021) [14]. The hair cortisol level in dogs was lately examined by van der Laan (2022) [15] in shelter dogs and by van Houtert in working dogs (2022) [16]. HCC is also used to assess stress in farm animals [17].

This study aimed to assess whether cats staying in an environment richer in resources (enriched environment) will be less stressed than those with fewer resources (standard environment). The research was carried out in three stages: 1. observations and collection of biological material from cats in a given shelter, 2. purchase and introduction of environmental enrichments and placing them in the same shelter, and 3. repeated observations and collection of biological material from cats in this shelter. We hypothesized that cats living in an enriched environment would have lower hair cortisol levels than those with fewer resources (standard environment).

## 2. Materials and Methods

The research was part of the project “Reducing the stress level in shelter cats through the use of environmental enrichments” under the program “Student Science Clubs Create Innovations” financed by the Ministry of Education and Science in Poland (contract No. SKN/SP/534344/2022). Approval for the study was obtained from the Animal Welfare Committee of the Faculty of Animal Sciences and Bioeconomy of the University of Life Sciences in Lublin, Poland (ZdsDZ/1/2022 of 25 May 2022).

Ten state shelters for homeless animals signed up to participate in the project. All shelters had a similar system for organizing the cats’ living environment. The cats were kept in group cat houses with access to outdoor aviaries (Figure 1). The cats had access to hiding places, beds, and scratching posts (Figure 2). They were fed wet food once a day and had constant access to dry food and water. Food was served in bowls. Litter boxes, both open and closed, were placed next to each other (Figure 3), with one litter box for several cats. The litter in the litter box varies, depending on the economic circumstances of the shelter. The cats had access to hiding places, beds, and scratching posts. We assessed the resources available for cats in given shelters and defined this as a standard environment. Then we introduced various environmental enrichments to the shelters, depending on demand. Our goal was to generally enrich the environment, i.e., increase the amount of cat resources available. So, we added what was least available in a given shelter. The purchased enrichments were mainly various types of scratching posts and climbing frames, litter boxes, and toys for interactive feeding. We wanted there to be much more of everything, i.e., it was not the quality but the quantity of resources that was increased. We defined this adjusted environment as an enriched environment.

One hundred and seventy-nine cats (74 females and 105 males) participated in the study. One hundred and sixty-three of the cats were neutered. All cats participating in the study were mixed breeds. Cats that were under quarantine were excluded. The first group consisted of cats living in an environment with fewer resources (standard environment)—103 cats (44 females and 59 males) from ten shelters. The second group consisted of cats living in an enriched environment—76 cats (30 females and 46 males) from the same ten shelters. The research material consisted of cat hair. Hair samples were taken from the lumbar-sacral area using scissors in a non-invasive way; the first 1 cm of hair closest to the skin was used for analysis. In our study, we cut and used the first 1 cm of hair closest to the skin for analysis because firstly, cat hair grows at a rate of 1 cm per month, and secondly, some studies indicate that the further the hair is from the skin, the lower the cortisol concentration may be. Kirchbaum called this the washout effect [18]. In cats, this may be related to the frequency of self-grooming [19]. We took 1 cm to determine cortisol in the last month and the highest cortisol concentration section. Since it is assumed that hair grows an average of 1 cm per month [20,21], each cat stayed in the shelter for at least one month and a maximum of two months before collecting biological material. This applies to both the first and second parts of the research. In the first stage, we collected hair from cats that had been living in a standard environment for at least one month. After adding environmental enrichments, we waited at least one month before collecting hair a second time. The cats in the second stage of the research were not the same as those in the first stage, so we didn’t use the “shave-reshave” method [22,23].

Hair was collected only from cats socialized with humans, showing no fear or aggression towards humans. Until biochemical analyses, the material was stored in string bags in a dry place at room temperature, adequately described. More cats used the resources than participated in the research.

Extraction methodology was taken from Koren et al. [24] and Accorsi et al. [25]. Hair was first minced into 1–2 mm length fragments, and 20 mg of trimmed hair was put in a glass vial. A total of 3.5 mL of methanol (Sigma-Aldrich, Poznań, Poland) was added, and vials were incubated at 50 °C with gentle shaking for 24 h. After incubation, the supernatant was filtered to separate the liquid phase and put into disposable glass culture tubes. Next, this supernatant was evaporated to dryness under an air-stream suction hood at 37 °C. Dry residue was then dissolved into 1 mL of phosphate-buffered saline (PBS) 0.05 M, pH 7.5. Samples were vortexed for one minute, followed by another 30 s until they were well mixed. The cortisol concentrations in the samples were determined with the DRG Salivary Cortisol HS ELISA assay. The procedures followed the manufacturer’s instructions. Cortisol concentrations were expressed in ng/mg.

Because the distribution of cortisol levels significantly deviated from normality, the Mann–Whitney U rank test—Z statistic—was used to test the significance of differences between the assessed groups of cats (enriched and standard environments). A positional measure of the average value, i.e., the median and quartiles, were used to describe the distributions. For selected parameters, the distribution of the examined features in individual months was illustrated on a categorized box chart based on positional measures (median, quartiles). The compliance of the distributions of the examined features with the normal distribution was assessed using the Shapiro–Wilk test. Statistically significant results were considered to be those that were significant at a typical level of significance, i.e., when *p* < 0.05. Arithmetic means, standard error, and range are also provided for a more complete presentation of the research results.

## 3. Results

The average hair cortisol level of shelter cats from an environment with a standard amount of resources and an environment enriched with a large amount of new resources was compared. Lower cortisol levels were found in cats’ hair from an enriched environment (*mean 0.059 ng/mg*) than in a standard environment (*mean 0.101 ng/mg*). Descriptive statistics are given in Table 1. Assessment with the non-parametric Mann–Whitney U test showed a statistically significant difference between the compared groups (*Z* = *4.893*; *p* = *0.000001*) (Figure 4).

The average cortisol level in females in the standard environment was 0.101 ng/mg, while in males, it was 0.102 ng/mg; in an enriched environment, it was 0.062 ng/mg in females and 0.056 ng/mg in males. Differences in cortisol levels between sexes were not statistically significant.

## 4. Discussion

The main goal of our study was to check whether cats in an enriched shelter could have lower hair cortisol levels than cats from a poorer environment. Our results show that the research hypothesis may be true. The hair cortisol level was significantly lower in the group of cats from an enriched environment than in the group from a standard environment. Similarly, McCobb et al. [26] tested whether cats in a modern shelter with environmental enrichment would be less stressed than in a traditional, “resource-poor” shelter. Cats in shelters with traditional cat housing had a significantly higher urinary cortisol to creatinine ratio than cats in enriched shelters.

High levels of stress in cats can cause changes in food intake, grooming, general activity, exploratory behavior, facial marking, and interactions with other cats and humans, as well as increased vocalization, anxiety, urine spraying, and aggressive behavior [13,27]. High levels of stress also affect the physical health of cats. Prolonged exposure to cortisol can have a toxic effect on areas of the brain involved in memory and lead to a loss of prior ability to deal with stressors [28].

Keeping shelter cats in a group does not have to be a stressor. In our research, we did not check whether cortisol levels were related to the density of the cat population in a given shelter. The shelters participating in the study were overcrowded with cats. The quantity does not change over time; instead, there is rotation. Studies on other animal species show that the population density in a group of captive animals significantly affects HCCs, with greater density associated with higher cortisol levels [29]. In the study by Uetake et al., the ratio of cortisol to creatinine in the urine of cats housed individually was higher than in cats housed in groups. Their research further indicated that cats become less active when housed individually in cages [30]. No significant difference was found in stress scores between cats from single-cat households and those from multiple-cat households [31]. In Wojtaś’s study, similarly, the number of cats living in the household did not affect the hair cortisol level of these animals [13]. An insufficient amount of resources in the environment may generate conflicts among cats but does not necessarily cause an increase in the frequency of aggressive behavior in cats [32].

The impact of various types of environmental enrichment on cats has already been the subject of many studies. So far, among other things, it has been described that hiding enrichment reduces behavioral stress in shelters [33,34,35]. Enrichment, such as visual stimulation, may have some enriching potential for domestic cats in shelters [36]. However, in a pilot study by Tuozzi et al. [37], cats that could see humans but did not have direct physical contact often vocalized and scratched intensively at the door. In Vitale & Shreve’s research, cats preferred contact with humans over toys or scented enrichments [38].

Another type of environmental enrichment may be olfactory enrichment. Ellis and Well [39] suggest that certain scents, especially catnip, can potentially enrich the environment for domestic cats. Interestingly, it turns out that cats use substances contained in catnip as a mosquito repellent [40]. Chadwin et al. [41], in turn, checked the effect of synthetic pheromones on shelter cats but found no evidence that the synthetic equivalent of cat cheek pheromones had any effect on the stress level in cats staying in shelters. Cognitive enrichment can positively impact shelter cats showing frustration [42]. Cats exposed to clicker training showed significantly more exploratory behavior and reduced inactivity [43]. In our study, we enriched each shelter with the resources that, in our opinion, cats lacked the most. Cats have many unique behaviors and needs, such as the need to hide, scratch, and obtain food by hunting, and thoroughly examining their environment is crucial to developing an effective plan to correct deficiencies [44].

Stress and the stress reaction are very complex processes. Measuring a single stress hormone may sometimes result in drawing incorrect conclusions. We did not consider the cortisol levels in individuals because that would involve the need to analyze their sex, age, physiological state, health status, etc. Our goal was to assess the mean cortisol level in individual research groups. However, the cats participating in stage 1 of the study (standard environment) had been adopted before we collected hair in stage 2 (enrichment environment). Thus, the cats from the second stage of the research are not the same as those in the first stage of the research. This is one of the limitations of our research. The literature reports that for the use of HCC as a biomarker of stress, it is important that the sample contains enough actively growing hairs, which can be achieved by the “shave-reshave” method [12,22,23]. Therefore, it could be better to have two samples of the same animal to make sure that cortisol levels refer to exactly that period.

## 5. Conclusions

The study aimed to assess the long-term stress level of cats in animal shelters with a standard amount of resources and in shelters with a significantly enriched environment. Our research results indicate that cats from a more enriched environment have significantly lower cortisol levels in their hair. Therefore, the role of resources in the cat environment, especially in shelters for homeless animals, could be significant and cannot be ignored when planning, organizing, or modernizing this type of environment.

## Figures and Tables

**Figure 1 animals-14-01392-f001:**
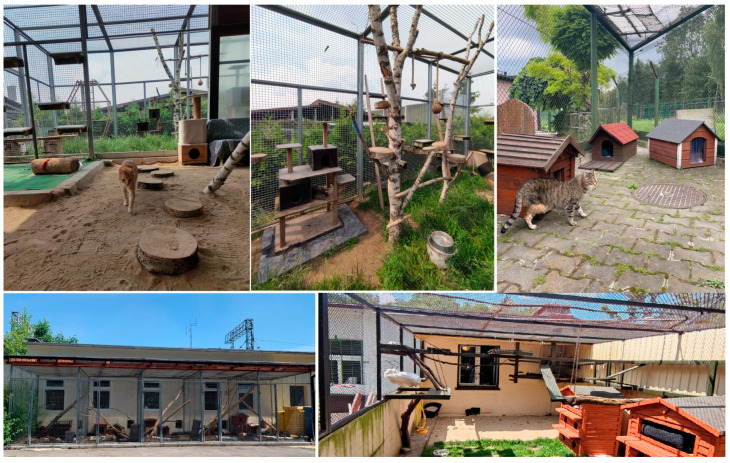
Examples of outdoor aviaries available for cats.

**Figure 2 animals-14-01392-f002:**
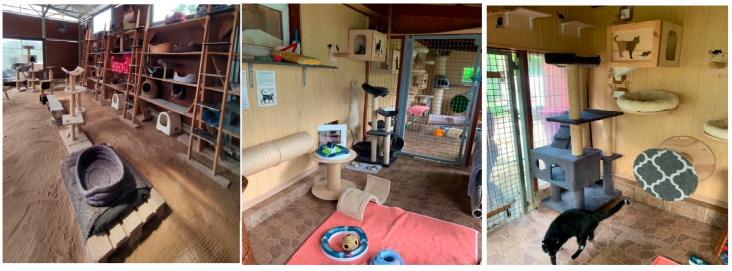
Examples of hiding places, beds, and scratching posts.

**Figure 3 animals-14-01392-f003:**
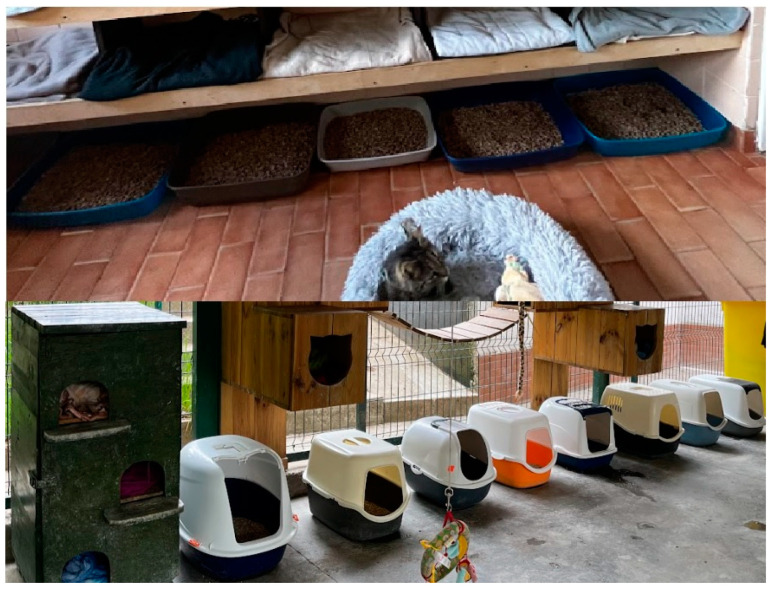
Examples of litter boxes.

**Figure 4 animals-14-01392-f004:**
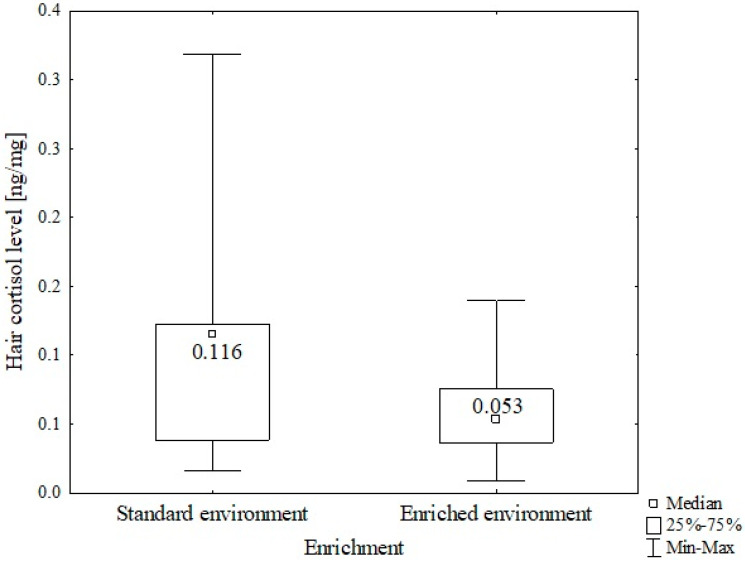
Comparison of the average hair cortisol levels in cats depending on environmental resources. U Mann–Whitney test *Z* = *4.893*; *p* = *0.000001*.

**Table 1 animals-14-01392-t001:** Values of the hair cortisol levels (ng/mg).

Environment	Mean ± SE	N	Range	Median	Q25–Q75	*p*-Value
standard	0.101 ± 0.006	103	0.016–0.319	0.116	0.039–0.126	0.000001
enriched	0.059 ± 0.003	76	0.009–0.140	0.053	0.037–0.075	

## Data Availability

The datasets generated during and/or analyzed during the current study are available from the corresponding author upon reasonable request.

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
