# Peer review of "The Impact of Environmental Enrichment on the Cortisol Level of Shelter Cats"

_animals, 2024, doi:10.3390/ani14091392_

Round 1

Reviewer 1 Report

Comments and Suggestions for Authors

This manuscript describes the use of hair cortisol concentrations (HCCs) to evaluate the influence of environmental enrichment on long-term adrenocortical activity and chronic stress in shelter cats. The results suggest that enrichment reduces HCCs, which is potentially important information for shelter management. However, there are several aspects of the manuscript that require significant revision by the authors.

INTRODUCTION

1. The Introduction describes the types of environmental enrichment developed for cats. It also briefly introduces the use of HCCs for assessing long-term cortisol output. However, this section should be expanded with additional information on how the hair cortisol approach has been applied in studies of feline stress (e.g., Wojtas, J., Journal of Feline Medicine and Surgery, 25(2), 2023, https://doi.org/10.1177/1098612X221150624; Contreras et al., Journal of Veterinary Internal Medicine, 35:2662–2672, 2021, DOI: 10.1111/jvim.16283). Another relevant citation is van der Laan et al., Scientific Reports, 12:5117, 2022, https://doi.org/10.1038/s41598-022-09140-w, which describes the use of HCCs as an index of stress in shelter-housed dogs.

2. The authors inappropriately placed the aim, hypothesis, and general methodological approach in the Materials and Methods section. Please move this material to the end of the Introduction, which is where it belongs.

MATERIALS AND METHODS

3. Key elements of the method are inadequately described. The authors state that they "assessed the resources available for cats in given shelters" and that they "introduced various environmental enrichments to the shelters, depending on demand." They then evaluated the influence of enrichment on HCCs. What's unclear is whether this is a cross-sectional study or a longitudinal study, which, among other things, is relevant for the statistical analyses. That is, did you divide the shelters into "enriched" vs. "standard" and then compare them? In that case, you need to indicate how many shelters were in each group. You also need to be clearer about how you evaluated "standard environment" vs. "enriched environment." Was this just a subjective evaluation, or did you use some method to quantify the amount of enrichment in the environment? Alternatively, did you collect hair samples from all shelters under the standard conditions (baseline samples) and then collect a second set of samples AFTER adding the environmental enrichment (enrichment samples)? If so, how long did you wait before collecting the second set of hair samples? Also in this case, did you perform statistical tests using a matched sample approach (since samples 1 and 2 will be correlated with each other)?

3. Research with rhesus monkeys has shown that the population density in a group of captive animals has a significant effect on HCCs, with greater density associated with higher cortisol levels (Dettmer et al., Psychoneuroendocrinology, 42, 59-67, 2014, http://dx.doi.org/10.1016/j.psyneuen.2014.01.002). Did you evaluate the population density (i.e,. number of cats per square foot of room space) in each shelter and determine whether HCCs were affected by density?

4. The authors stated that the hair in their cats grew at an average rate of 1 cm per month, but they provide no citation for this information. I suggest using the following: Baker KP. Hair growth and replacement in the cat. Br Vet J. 1974; 130:327-335.

5. The authors report the HCC values in ng/ml, which is concentration in a liquid. However, hair is obviously a solid, so therefore the assay output for each sample needs to be converted to the amount of cortisol PER UNIT WEIGHT HAIR. Assuming that every hair sample weighed EXACTLY 20 mg, then the following simple equation can be used for this conversion: measured cortisol concentration in ng/mL x 3.5 (ratio of the volume of methanol used in the extraction to the volume of buffer used to reconstitute the sample) divided by 20 (hair sample weight). The resulting units are ng cortisol per mg hair (ng/mg). If individual samples had different weights, then you need to substitute that specific weight (in mg) for the "20" value used in the above equation.

RESULTS

6. The HCC values shown in Fig. 4 box-and-whisker plots (i.e., 0.66 vs. 0.31) differ from both the mean and median values shown in Table 1. Please check this and correct it as appropriate so that the numbers agree.

TYPOS/MINOR ERRORS

7. In the second line from the end of the last paragraph of the Introduction, please change "HCL" to "HCC".

8. In the part of the Materials and Methods that mentions the volume of methanol used to extract each hair sample, please add "mL" after "Three and a half".

Reviewer 2 Report

Comments and Suggestions for Authors

Overview and general recommendation:

This manuscript is a communication as a part of a bigger project titled “Reducing the stress level in shelter cats through the use of environmental enrichments". The topic is very important regarding the welfare of cat, especially in an environment which puts it to the test. The study aimed to compare the hair cortisol level of cats living in enrichment environment to cats living in a non-enriched one. The paper covers a topic in the field of cat welfare, which is suitable for this journal, and refers to most relevant literature in the area. 

The methodology is not really clear, and there is an issue regarding the hair collection. I suggest that this part be made clearer and implemented.

Specific comments are provided below.

Specific comments

Abstract

It would be better to add in the abstract the presence of the two group of cats.

Introduction

-Lines 28-30: please add a reference for this statement.

-Lines 46-47: please add a reference for this statement.

-The aim is missing: please move the sentence present in “Materials and Methods” (lines 60-62) in this section.

Materials and Methods

-Please explain better the presence of the two groups of shelters (one with enrichment and the other one without).

-It could be better explaining exactly which enrichment you add. it would be correct to explain which base was considered unenriched and which was enriched: indicating the final appearance of the enriched environment. You could describe better the two groups regarding the differences.

-Lines 85-87: you state that the number of cats was 179 and that “Cats that were under quarantine were excluded”, but these cats were out of the 179? 

-Lines 87-89: From how many shelters do the cats in the first group come and from how many in the second group? 

-Line 89: replace the dot with a semi-colon after “way”.

-Lines 96-97: “Hair was collected only from cats socialized with humans, showing no fear or aggression towards humans”: so the number was 179 or less?

-Lines 93-95: Is not clear for me why you add this statement, could you explain the importance or the relation with your operation?

-Another question regards the hair collection: 

You state “Since it is assumed that hair grows an average of 1 cm per 91 month, each cat stayed in the shelter for at least one month and a maximum of two months 92 before collecting biological material”, so you collected just one sample for each cat, right? In literature is reported that for the use of HCC as a biomarker of stress, it is important that the sample contains enough actively growing hairs, which can be achieved by the “shave-reshave” method. A certain area is shaved at the beginning of the period of interest, and the regrown hair in the same area is reshaved at the end of this period (normally one month) (Heimbürge et al., 2019; Davenport et al., 2008; Meyer and Novak, 2012).

Discussion

Lines 141-144: please use a more probabilistic tone given the issues reported in the materials and methods section. Please use the same tone in the conclusion

The discussion is not clearly really related on your work, please connect the statement reported in the discussion with your work of enrichment and with the environment present in your shelters. 

Reviewer 3 Report

Comments and Suggestions for Authors

This paper is a study that examined whether cortisol, a marker of stress, differs depending on the enrichment of shelter cats. Feral cats are temporarily sheltered in shelters, and enrichment is a very important issue in terms of feline health. This study has a social significance because it shows that cats in enriched shelters tend to have lower cortisol levels. On the other hand, the lack of a detailed description of the level of enrichment in the shelters studied and the lack of a discussion of hormones and stress behavior may need to be corrected. I point out some points below.

Major points

Please create a diagram or table that objectively shows the facilities of POOR enrichment shelters and RICH shelters.

Why did you not compare PRE and POST when introducing furniture, etc.?

According to the emotional reactivity hypothesis, increased domestication would decrease the stress response and lower cortisol. Please provide detailed information about the cats, such as the percentage of mongrel/pedigreed cats between the two shelters.

Cortisol increases with increased exercise, please discuss this point.

This study focuses only on cortisol. What do you think is the relationship to stress behavior? For example, McCobb's study found no correlation between cortisol and stress behavior. We think it would be better if you could organize that in your discussion for the reader.

Minor points

L96 Please tell us how many cats there were in each shelter overall and how many of them did not shed and how many did!

Result Please italicize the statistics

L147 I don't think the p-value is necessary

L161 Ellis, 2008 should be replaced with a number.

L158 paragraph What is the most effective enrichment currently conceivable? Please add a summary sentence for this paragraph.

L168 Catnip may be effective because catnip has no dependencies. You may also cite Uenoyama's paper.

Uenoyama et al (2023)

Assessing the safety and suitability of using silver vine as an olfactory enrichment for cats

https://doi.org/10.1016/j.isci.2023.107848

L174 What is unique behavior? Please be specific.

Round 2

Reviewer 1 Report

Comments and Suggestions for Authors

In this revised version, the authors have improved the manuscript. However, several problems remain that require clarification of the methodology and the results, and updating of the References section.

1. In their response to reviewers, the authors indicate that the cats that participated in the first phase (initial hair collection) were all adopted out before the second phase of the study (second hair collection after enrichment). This is STILL NOT CLEAR from reading the Methods section. In fact, the sentence on lines 104-105 of page 4 ("After adding environmental enrichments, we waited at least one month before collecting hair), made me think that the same cats were used for both the first and second hair sampling. Please add a statement indicating that the cats participating in phase 1 (standard environment) were adopted before phase 2 and that a different group of cats entered the shelters and contributed hair cortisol in phase 2 (enrichment environment).

2. Please break down the number of males and females in each group (standard vs. enriched environment), not just the overall sex distribution.

3. The authors state that they still wish to use ng/ml as the metric for hair cortisol since that's what they did in previous publications. The fact is that this approach is wrong, regardless of whether you used it before. For example, see the Contreras et al. paper I recommended for inclusion in the Introduction. Those authors used pg cortisol per mg hair as their unit of measure. Likewise, the VAST MAJORITY of hair cortisol papers in all species correct their assay output to amount of cortisol per unit weight of hair. Please perform the mathematical conversion of each cortisol value to correct for hair weight, recalculate the means, medians, etc., redo the statistical analyses, and then show the converted data in the text, tables, and figure.

4. Lastly, the authors made some changes in the references, but this was done in a very sloppy way. (a) In the text, the "Iqubal" citation (p. 2, line 48) doesn't include the year or the number of the reference in the Reference section. (b) The same is true for the Wojtas and Contreras references (p. 2, line 53), where no year is given for Wojtas et al. and neither citation includes a number in the Reference section. (c) I mentioned in my previous review that it would be of interest to readers to cite the van der Laan et al., 2022 paper on hair cortisol in shelter dogs. In the text (p. 2, line 53), this citation is incorrectly put in with the cat citations. More appropriate would be to mention in the Discussion what those authors found in their dog shelter study and to compare/contrast those results with the present results in cats. (d) The citation for hair growth rate in cats (Baker, 1974) (p. 4, line 101) is not listed, but instead the authors list reference #13 which is for a different paper. (e) Lastly, it appears that the authors failed to add the new references to the Reference section itself. Please make sure that every citation in the ext of the manuscript exactly matches the corresponding reference in the Reference section.

Reviewer 2 Report

Comments and Suggestions for Authors

Overview and general recommendation:

The authors did a good job addressing my previous comments. I have just a few more remarks

Specific comments

Materials and Methods

-Lines 93-95: Is not clear for me why you add this statement, could you explain the importance or the relation with your operation? In our study, we cut and used 1 cm of hair closer to the skin for analysis. At this point in the text, we wanted to explain why it is essential that only this 1 cm and only as close to the skin as possible. Because firstly, the hair grows 1 cm per month, and secondly (and this is what you are asking), some studies indicate that the further the hair is from the skin, the lower the cortisol concentration may be. Kirchbaum called this the washout effect. In cats, it may be related to self-grooming. We took 1 cm to determine cortisol in the last month and in the highest cortisol concentration section. 

Thanks for your reply, if you could you please explain like this in the test, it would be clearer

-Another question regards the hair collection:
You state “Since it is assumed that hair grows an average of 1 cm per month, each cat stayed in the shelter for at least one month and a maximum of two months before collecting biological material”, so you collected just one sample for each cat, right? In literature is reported that for the use of HCC as a biomarker of stress, it is important that the sample contains enough actively growing hairs, which can be achieved by the “shave-reshave” method. A certain area is shaved at the beginning of the period of interest, and the regrown hair in the same area is reshaved at the end of this period (normally one month) (Heimbürge et al., 2019; Davenport et al., 2008; Meyer and Novak, 2012).
Yes, we collected just one sample for each cat. The cats participating in stage 1 of the study were (luckily) adopted before we collected hair in stage 2. The cats from the second stage of research are not the same cats as in the first stage of research.

Since in literature is reported that for the use of HCC as a biomarker of stress, it is important that the sample contains enough actively growing hairs, which can be achieved by the “shave-reshave” method, so it could be better have 2 sample of the same animal to make sure that cortisol levels refer to exactly that period (Heimbürge et al., 2019; Davenport et al., 2008; Meyer and Novak, 2012), It would be better add these aspect in your discussion as a possible limit of the research.

Discussion

-As reported above, please add the discussion on HCC sampling.

-The discussion is not clearly really related on your work, please connect the statement reported in the discussion with your work of enrichment and with the environment present in your shelters. The topics discussed in the discussion and the works cited most closely refer to our results or are indirectly related to them. As suggested by other reviewers, we just try not to repeat our results in the discussion. 

The changes that you made improved the discussion, now I’m happy with that.

Round 3

Reviewer 1 Report

Comments and Suggestions for Authors

The authors have responded to my concerns and I have no remaining issues with the manuscript.